

# Effect of shooting experience on executive function: differences between experts and novices

Mingming Shao[1,2,*], Yinghui Lai[1,3,*], AnMin Gong[4], Yuan Yang[5], Tingting Chen[5] and Changhao Jiang[1]

[1] Beijing Key Laboratory of Physical Fitness Evaluation and Technical Analysis, Capital University of Physical Education and Sports, Beijing, China
[2] Faculty of Social and Public Administration, Guangdong Baiyun University, Guangzhou, China
[3] School of Education Science, Hunan Institute of Science and Technology, Yueyang, China
[4] School of Information Engineering, Engineering University of Armed Police Force, Xi'an, China
[5] College of Physical Education and Sports, Beijing Normal University, Beijing, China
[*] These authors contributed equally to this work.

## ABSTRACT

**Background**. Executive function (EF) plays an important role in controlling human actions. Shooting is a closed motor skill, characterized by high anti-interference ability and high mental intensity. However, the beneficial effects of closed exercises such as shooting on EF remain unclear.

**Methods**. We utilized an expert-novice paradigm and the Flanker task to examine the EF of shooting athletes. Participants were assigned into the novice group, expert group, or control group, based on the level of training and competition experience. Reaction time (RT) and accuracy of performance across the three groups were compared.

**Results**. For the simple task, the control group showed a longer RT than the novice group, for all three inter-stimulus interval (ISI) conditions. Significant differences between the control and the expert groups were observed only at 300-ms and 400-ms ISIs. For the complex task, the control group exhibited a higher RT than the novice and expert groups at the 300- and 400-ms ISIs.

**Conclusions**. The self-control during performing closed motor skills in the environment determines that shooters have proficient anti-interference ability. This ability is uncorrelated with task type, but most likely is affected by reserved task response time.

## INTRODUCTION

Executive function (EF) is an advanced cognitive function, through which the brain coordinates all available resources and controls the processes required to accomplish the specific goals of complex cognitive tasks (*Perner & Lang, 1999*). EF includes the processes of inhibition, working memory, cognitive flexibility, and planning. EF processes have different developmental trajectories, and the intensive development of each process begins at different periods. Each component is expected to contribute uniquely to an individual's abilities to resolve peer conflict competently, through the inhibition of incompetent social

Corresponding authors
Tingting Chen, ctt1226@163.com
Changhao Jiang, jiangchanghao@cupes.edu.cn

responses (response inhibition), the maintenance of social goals (working memory), and attentional shifts between complex social rules and potential response options (cognitive flexibility). Numerous studies have shown that exercise can have significant positive impacts on EF.

Motor skills can be categorized into open and closed (*Dai et al., 2013*; *Tsai & Wang, 2015*). Open motor skills, such as table tennis, basketball, and football refer to skills that can be altered by changes in the individual's external context and that require the individual to process external information and predict upco ming events (*Russo et al., 2010*). In a longitudinal study, *Kida, Oda & Matsumura (2005)* used a go/no-go paradigm to demonstrate that 2 years of baseball practice improved EF in college students. In some cross-sectional studies, *Nakamoto & Mori (2008)* reported that the reaction time (RT) of basketball and baseball players, in simple RT and go/no-go RT tasks, were shorter than those of non-players. *Furley & Memmert (2012)* showed that high-level basketball players can better resist external stimuli interference, focus on the current task, and make reasonable tactical decisions than lower-level basketball players. These studies have shown that open motor skills can have positive impacts on EF.

EF allows athletes to resist strong internal tendencies and external temptations; control attention, behavior, thinking, and emotions; and focus on the present to make appropriate behavioral decisions (*Diamond, 2013*). Closed motor skills refers to skills that can be performed without reference by changes in external conditions, such as yoga, shooting, and swimming. Also, closed motor skills generally have fairly fixed patterns of action. Recent reviews have documented the beneficial effects of closed motor skills on EF (*Bowden et al., 2011*). A randomized, controlled trial revealed that an acute bout of Hatha yoga resulted in significant improvements in working memory and selective attention, based on the Digit Letter Substitution task (*Telles et al., 2012*). *Maeshima et al. (2017)* showed that synchronized swimming has beneficial effects on cognitive function, particularly with regards to recent memory. Players focused on closed (or open) motor skills require the ability to quickly suppress their motor responses and make new decisions.

To ensure a normal level of performance, athletes who participate in shooting sports must be tightly engaged with the target when aiming and triggering the shooting gun/riffle ("attention control") (*Sattlecker et al., 2014*; *Ihalainen et al., 2015*). To achieve desired results, athletes must continuously monitor their behaviors to ensure that they meet the requirements of the sport. However, mistakes or errors may be made during practice and competition and it is important to minimize their effects on the outcome. After an error occurs, the athlete should promptly and accurately identify the problem ("emotion control") and make appropriate adjustments to ensure future accurate performance ("behavior control"). For example, during a skeet shooting competition, the athlete must pay attention to the direction the appearing target and its trajectory, and be able to adjust his/her shooting posture to achieve a successful hit. Emotion control and behavioral control enable the athletes to make quick and effective decisions to overcome negative impacts caused by previous errors or poor performances.

Shooting is a closed motor skills, characterized by high anti-interference ability and high mental intensity. The shooting movement has a particularly high requirement for executive

and restraining functions. In order to achieve the target performance during a competition, the athlete often adjusts his/her current behavior based on previous experience. Similar to the conflict adaptation effect (CAE) of the Flanker task, this cognitive behavior refers to the phenomenon that an individual can better solve conflicts in the process of human cognitive control if he/she experiences similar conflicts later. Therefore, this study adopted the classic paradigm of the Flanker task and hoped to recognize the relationship between shooting practice and EF through this paradigm. Participants with three levels of shooting experience were recruited: college graduate students without shooting experience (control group), shooting athletes with limited training and competition experience (novice group), and shooting athletes who had participated in international competitions or been among the top players in national competitions (expert group). We hypothesized that the reaction time (RT) would not differ among the three groups for performance of consistency or simple tasks; but for performing inconsistency or complex tasks, the expert group would exhibit the highest accuracy and fastest RT, followed by the novice group and control group.

## METHODS

### Participants

Thirty-two (32) right-handed participants were recruited, based on their shooting experience. The control group included 12 graduate students (mean age = 23.50 ± 0.34 years, 9 males), who had 0 year shooting experience. The novice group consisted of 9 athletes (mean age = 17.89 ± 1.15 years, 6 males), who had on average 2.84 ± 0.57 training and competition experience. The expert group contained 11 athletes on the Chinese national Clay Pigeon shooting team (mean age = 29.36 ± .71 years, 9 males), who had participated in international competitions or were among the top achievers in past national competitions (see Table 1). On average, the athletes in the expert group had 18.13 ± 2.48 years of training experience at the time of the study. All subjects had normal or corrected-to-normal visual acuity and were right-handed. No individuals reported any history of neurological, cardiovascular or musculoskeletal disorders, nor did any report taking medications that might affect cognitive or neuromuscular functions. The Institutional Ethics Committee of the Capital University of Physical Education and Sports approved the study, and all participants signed an informed consent before joining the study. The participants were instructed to refrain from consuming caffeine and alcohol beverages and to get at least 8 h of sleep the day before the experiment.

### Procedures

The Flanker task was run by using the E-prime program on a laptop computer, with a 14-in display screen and a resolution matrix of 1,024 ×768. The background color was black. The fixation point, "+" and the stimulus were white and were located at the center of the screen. The sizes of the fixation point and the stimuli were 0.6 cm and 29 cm ×10 cm, respectively. The target stimulus consisted of five arrowheads. Two categories of stimuli were established: consistent (>>>>>, <<<<<) and inconsistent (<<><<, >><>>) (see Fig. 1). The participants were asked to judge the orientation of the middle arrowhead and
Table 1 **Descriptive statistics for behavioral data.** Basic information about participants, such as gender and age stratification.

| Group | Age (years) | Shooting age (years) | Sex | |
|---|---|---|---|---|
| | | | Male | Female |
| Control group | 23.50 ± 0.34 | 0 | 9 | 3 |
| Novice group | 17.89 ± 1.15 | 3.39 ± 1.04 | 6 | 3 |
| Expert group | 29.36 ± 2.71 | 14.27 ± 2.66 | 9 | 2 |

(A)                                    (B)

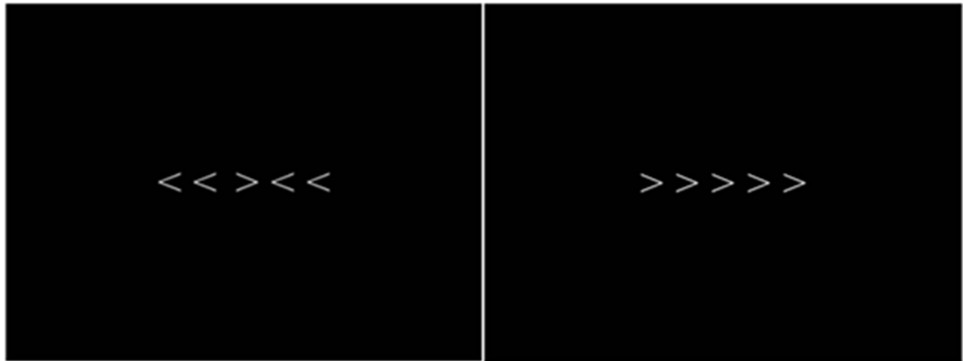

Figure 1 **An example of stimulus.** The flanking arrows either all pointed in the same direction as the target arrow (e.g., " < < < < <"), or they all pointed in the opposite direction (e.g., " < > < <"). The trials on which the flanking arrows pointed in the same direction as the target arrow were the congruent trials (A); the trials in which they pointed in the opposite direction were the incongruent trials (B). Subjects were to press "F" button for a left facing central arrow and "J" button for a right facing central arrow.

to ignore the arrowheads flanking the target. If the middle arrowhead was oriented to the left, the participant used their left index finger to press the F key. If the middle arrowhead was oriented to the right, the participant used the right index finger to press the J key. The participants was required to maintain the fingers on the response buttons (F and J keys) during the entire Flanker task. The experiment was performed in a quiet and well-lit laboratory.

The experiment adopted a reversal design (ABAB design) and was divided into one training module and four experimental modules. During the experimental modules, the first and third modules included consistent or simple tasks, whereas the second and fourth modules had inconsistent or complex tasks. Each experimental module comprised 36 trials over 30 s, and the four modules together comprised 144 trials over 120 s. The tasks required each participant to judge orientation of the middle arrows as quickly and accurately as possible. A rest period of at least 30 s was provided between two experimental modules to prevent the influence of the previous experimental module on the following module. Each experimental module began with a 500-ms introduction period, with the fixation point located on the center of the screen. The target stimulus was then presented for 150 ms,

**Table 2 Descriptive statistics for reaction time (ms) data.** Mean Reaction time (ms) with standard errors for each condition.

| Group accuracy | Consistent task | | | Inconsistent task | | |
|---|---|---|---|---|---|---|
| | 200 ms | 300 ms | 400 ms | 200 ms | 300 ms | 400 ms |
| Control group | 0.91 ± 0.12 | 0.93 ± 0.06 | 0.94 ± 0.08 | 0.74 ± 0.19 | 0.91 ± 0.07 | 0.85 ± 0.18 |
| Novice group | 0.91 ± 0.09 | 0.94 ± 0.06 | 0.95 ± 0.06 | 0.59 ± 0.32 | 0.63 ± 0.31 | 0.71 ± 0.22 |
| Expert group | 0.91 ± 0.08 | 0.96 ± 0.05 | 0.90 ± 0.09 | 0.80 ± 0.16 | 0.82 ± 0.12 | 0.88 ± 0.12 |
| Total | 0.91 ± 0.09 | 0.94 ± 0.06 | 0.93 ± 0.08 | 0.71 ± 0.24 | 0.79 ± 0.22 | 0.82 ± 0.19 |

**Table 3 Descriptive statistics for accuracy data.** Mean accuracy rate with standard errors for each condition.

| Group RT(ms) | Consistent task | | | Inconsistent task | | |
|---|---|---|---|---|---|---|
| | 200 ms | 300 ms | 400 ms | 200 ms | 300 ms | 400 ms |
| Control group | 161.34 ± 10.03 | 206.74 ± 36.18 | 227.92 ± 49.46 | 165.26 ± 19.39 | 210.70 ± 25.93 | 239.78 ± 43.29 |
| Novice group | 148.11 ± 8.41 | 165.50 ± 11.97 | 169.05 ± 19.38 | 152.39 ± 18.33 | 175.82 ± 31.65 | 195.15 ± 25.68 |
| Expert group | 153.63 ± 11.21 | 173.39 ± 16.37 | 172.97 ± 20.45 | 157.32 ± 7.11 | 183.05 ± 15.85 | 192.28 ± 20.63 |
| Total | 154.97 ± 11.16 | 183.68 ± 30.45 | 192.47 ± 43.36 | 158.91 ± 16.29 | 191.39 ± 28.59 | 210.90 ± 38.61 |

consistent or inconsistent. The interstimulus intervals (ISI) were 200 ms, 300 ms, and 400 ms. The participant was required to react as quickly as possible within the ISI. The RT was recorded as the time from the stimulus presentation to the time that the response button was pressed. Simple and complex stimuli with RTs less than 100 ms were excluded from the analysis as they were out of human RT range.

## Statistical analysis

The data were analyzed using a 3 (shooting experience: control group, novice group, expert group) ×2 (task type: consistent, inconsistent) ×3 (ISI: 200 ms, 300 ms, 400 ms) analysis of covariance (ANCOVA), in which shooting experience was a between-group variable, task type, and ISI were within-group variables, and age was a covariate. RT and accuracy were dependent variables. SPSS 23.0 statistical software was used to perform all statistical analyses. Partial eta-squared ($\eta^2$) was reported, to provide an overall association index for the proportion of total variance accounted for by any treatment effect. The value range for partial $\eta^2$ is 0.00–1.00. According to *Shieh (2015)*, a larger partial $\eta^2$ value reflects a stronger association between factors and dependent measures for empirical studies.

## RESULTS

The effects of shooting experience on the cognitive inhibition of EF are summarized in Table 1. Table 2 presents the RTs for the consistent and inconsistent tasks in the three groups, and Table 3 presents the accuracy results for the two tasks in the three groups.

### Reaction time

The RT data were subjected to 3 (shooting experience groups) ×2 (task types) ×3 (ISI) repeated measures ANCOVA (see Fig. 2, Table 4). The results showed that the main effect of shooting experience was significant [$F_{(2, 28)} = 9.441$, $p < 0.001$, $\eta^2 = 0.403$] and

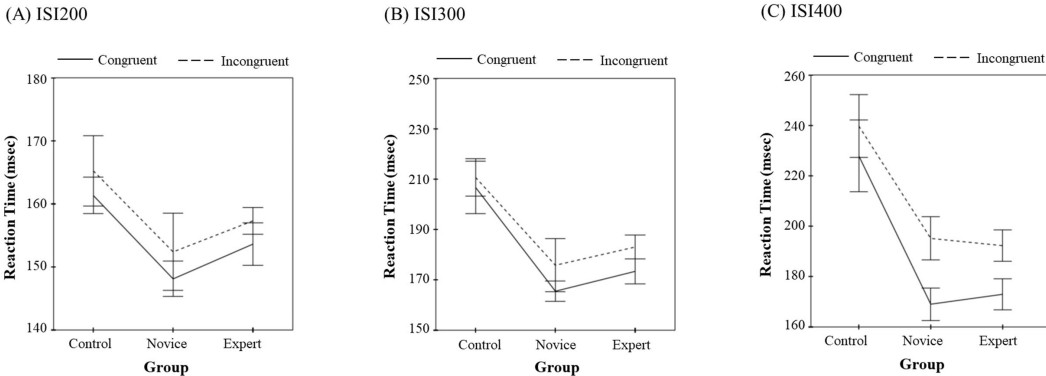

**Figure 2** **Reaction Time (ms).** Effects of the interaction between shooting experience and task type on RT by ISI 200 (A), ISI 300 (B), and ISI 400 (C).

**Table 4** **ANCOVA TEST(RT).**

|  | df | F | $p$ | $\eta^2$ |
|---|---|---|---|---|
| age | 1 | 0.422 | 0.521 | 0.015 |
| group | 2 | 9.441 | 0.001 | 0.403 |
| ISI | 2 | 1.837 | 0.169 | 0.062 |
| ISI * age | 2 | 0.397 | 0.674 | 0.014 |
| ISI * group | 4 | 8.280 | 0.000 | 0.372 |
| task | 1 | 2.140 | 0.155 | 0.071 |
| task * age | 1 | 0.246 | 0.624 | 0.009 |
| task * group | 2 | 0.688 | 0.511 | 0.047 |
| ISI * task | 2 | 3.552 | 0.035 | 0.113 |
| ISI * task * age | 2 | 1.946 | 0.152 | 0.065 |
| ISI * task * group | 4 | 0.512 | 0.727 | 0.035 |

that the interaction between shooting experience and ISI was significant [$F(4, 56) = 8.28$, $p < 0.001$, $\eta^2 = 0.372$]. The interaction between shooting experience and task type was also significant [$F (2, 56) = 3.552$, $p < 0.05$, $\eta^2 = 0.113$]. No other main effects or interactions reached significance.

For consistent trials, the pairwise comparison of groups with ISI showed that for the 200-ms ISI, a significant difference was observed between the novice and control groups ($p = 0.028$), with the control group showing longer RTs than the novice group. Differences among other groups were not significant. For the 300-ms ISI, significant differences were seen between the expert and control groups ($p = 0.015$) and between the novice and control groups ($p = 0.010$); the difference between the expert and novice groups was not significant, and the control group showed longer RTs than the expert and novice groups. For the 400-ms ISI, the results were similar to those for the 300-ms ISI condition.

For inconsistent trials, the pairwise comparison of groups with ISI showed that for the 200-ms ISI, no significant differences among groups were observed. For the 300-ms ISI, significant differences were observed between the expert and control groups ($p = 0.041$)

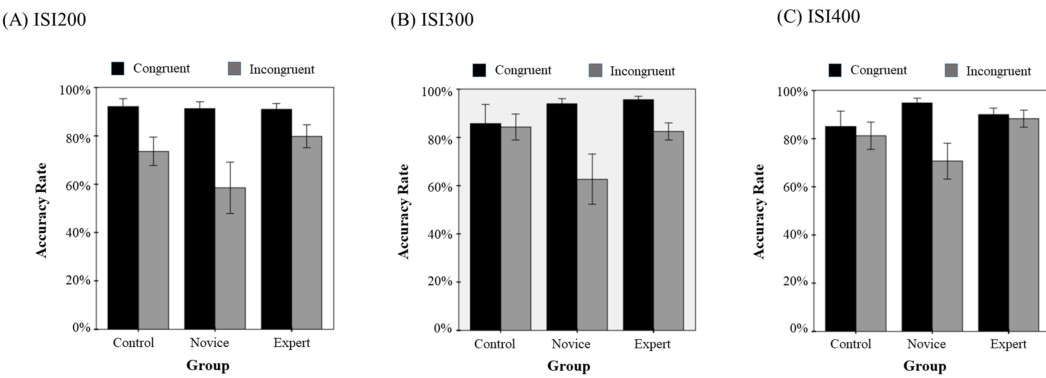

**Figure 3** **Accuracy Rate.** Effects of the interaction between shooting experience and task type on ACC by ISI 200 (A) ISI 300 (B) and ISI 400 (C).

and between the novice and control groups ($p = 0.037$), with the control group showing longer RTs than the expert and novice groups; no significant RT differences were seen between the expert and novice groups. For the 400-ms ISI, the results were similar to those for the 300-ms ISI condition.

### Accuracy

A 3 (shooting experience groups) ×2 (task types) ×3 (ISI) repeated measures ANCOVA for accuracy data revealed that only the main effect of task type was significant (Fig. 3; $F(1, 26) = 6.364$, $p < 0.05$, $\eta^2 = 0.197$), suggesting that the accuracy of the Flanker effect was not affected by both the shooting experience and ISI. No other main effects or interactions reached statistical significance.

## DISCUSSION

The results of behavioral experiments showed that for the simple task, both the novice and expert groups demonstrated faster RT than the control group for all three ISI conditions. For the complex task, the RT results were consistent with those observed for the simple task. However, these differences were not found in the accuracy measurement. Therefore, the improved behavioral responses of the novice and expert group compared with the control group were primarily due to shooting experience.

Our behavioral data do not fully support our research hypothesis. The accuracy recorded for all tasks is in line with our hypothesis. Previous open motor skills studies that examined the inhibitory control of athletes found that athletes, such as baseball and basketball players, responded faster or committed fewer errors compared to non-players (*Nakamoto & Mori, 2008*). The performance of a simple task only requires a quick instinctive response without a significant cognitive process. Our results indicate the absence of individual differences among the three groups. However, in another study of disabled athletes, comparisons among open-sport and closed-sport athletes and healthy non-athletes showed that, although the RT was slower in disabled participants, the accuracy was similar in all three groups (*Russo et al., 2010*). In addition, these studies provides support for the argument that

long-term training improves EF (*Cheng et al., 2017*; *Fronso et al., 2016*; *Kim et al., 2013*). *Chan et al. (2011)* reported that high-fitness fencers showed superior inhibitory control relative to non-athletes, whereas low-fitness fencers did not display improved inhibitory control. In contrast, *Wang et al. (2013)* reported that open motor skill athletes (tennis players) had shorter stop-signal RT compared with swimmers and sedentary controls, whereas no difference was found between the closed motor skill athletes (swimmers) and sedentary controls.

Additionally, the RT results were contrary to our hypothesis but instead conform to the assumptions of complex tasks. Our data did not match the results of previous studies, which reported that during simple consistency tasks, the RT of novice and expert tennis players did not differ (*Etnier et al., 2006*; *Kovacs, 2007*). The discrepancy between our findings and those of previous studies may be due to the use of Clay Pigeon shooters in our study. Clay Pigeon shooters must complete a series of actions, including gunning, aiming, firing, and shooting, within a 0.4–0.6-second period. Therefore, Clay Pigeon shooting requires a faster response among shooting athletes.

The results showed no significant difference between novice and expert groups for the simple task, which is consistent with previous studies. Clay Pigeon shooting has the shortest theoretical training period among all shooting sports. For example, the normal training period for men's pistol slow shooting is 10 years, whereas the average training period for Clay Pigeon shooting is only 2 years. In other words, with 2 years of hard training, individuals can improve their Clay Pigeon shooting performance to the professional level. The training experience of the shooters in the novice group in this study was at least 2 years. This result also confirmed the research hypothesis. For complex task, the RT of the expert group was longer than that of the control and novice groups. This result demonstrates that training can improve EF and is consistent with the findings of previous studies (*Kida, Oda & Matsumura, 2005*; *Nakamoto & Mori, 2008*).

This study demonstrates improved inhibitory function in closed motor skill athletes. According to previous research, open motor skills and closed motor skills have different effects on EF. Therefore, investigators should perform further research to examine dual sports at the longitudinal study.

## CONCLUSION

The self-control during performing closed motor skills in the environment determines that shooters have proficient anti-interference ability. This ability is not correlated with task type, but most likely is affected by reserved task response time. In the short ISI, participants in the three groups are impossible to quickly and accurately respond to the stimuli, which leads to a "floor effect" of behavioral performance. In contrast, the longer ISI reserves an adequate reaction time.

## ACKNOWLEDGEMENTS

We express our thanks for the help of the Beijing Shooting School and China National Shooting Team Coach Xiao Wang, and Professor Guanghui Yue from the Kessler

Rehabilitation Center of the United States. We thank the National Center for Protein Sciences at Peking University for assistance with data analysis.

### Funding

This work was supported by the National Natural Science Foundation (NNSF) of China under Grant (No. 31771244), Research Fund of Beijing Institute of Sports Science (2017) and Open Research Fund of the State Key Laboratory of Cognitive Neuroscience and Learning to Changhao Jiang, and Education work committee of the Beijing committee of the communist party of China [grant number 2016000020124G094]; and Beijing education committee [grant number SM201810029001] and "13th Five-Year Plan Project" of Philosophy and Social Sciences, Guangdong Province [grant number GD19CXL01]. The funders had no role in study design, data collection and analysis, decision to publish, or preparation of the manuscript.

### Grant Disclosures

The following grant information was disclosed by the authors:
National Natural Science Foundation (NNSF) of China: 31771244.
Research Fund of Beijing Institute of Sports Science (2017).
Open Research Fund of the State Key Laboratory of Cognitive Neuroscience.
Learning to Changhao Jiang, and Education work committee of the Beijing committee: 2016000020124G094.
Beijing education committee: SM201810029001.
13th Five-Year Plan Project" of Philosophy and Social Sciences, Guangdong Province: GD19CXL01.

### Competing Interests

The authors declare there are no competing interests.

### Author Contributions

- Mingming Shao conceived and designed the experiments, performed the experiments, analyzed the data, prepared figures and/or tables, authored or reviewed drafts of the paper, and approved the final draft.
- Yinghui Lai conceived and designed the experiments, analyzed the data, prepared figures and/or tables, authored or reviewed drafts of the paper, and approved the final draft.
- AnMin Gong and Yuan Yang performed the experiments, authored or reviewed drafts of the paper, and approved the final draft.
- Tingting Chen analyzed the data, prepared figures and/or tables, and approved the final draft.
- Changhao Jiang conceived and designed the experiments, authored or reviewed drafts of the paper, and approved the final draft.

## Human Ethics

The following information was supplied relating to ethical approvals (i.e., approving body and any reference numbers):

The Capital University of Physical Education and Sports granted Ethical approval to carry out the study within its facilities.

## Data Availability

The raw measurements are available in the Supplementary Files.

## Supplemental Information

Supplemental information for this article can be found online at http://dx.doi.org/10.7717/peerj.9802#supplemental-information.

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
