# Peer review of "Effect of shooting experience on executive function: differences between experts and novices"

_PeerJ, doi:10.7717/peerj.9802_

## Round 0.1 · original submission · Major Revisions

I now have received two reviewers' comments. Although both reviewers expressed their interest in your study, several aspects of this manuscript should be revised to improve its clarity. Their observations are presented with clarity so I'll not risk confusing matters by belaboring or reiterating their comments. While I might quibble with the occasional point, I note that I regard the reviewers' opinions as substantive and well-informed. I believe that all of the highlighted reservations require contemplation and appropriate attention in revising the document if it is to contribute appropriately to Peerj and the extant literature. Please revise or refute according to the two reviewers' comments and provide a point by point reply in addition to the revised manuscript.

Tsung-Min Hung, PhD., FNAK, FISSP
PeerJ editor
Research chair professor,
Department of Physical Education,
National Taiwan Normal University

·

Basic reporting

The manuscript reports a result of the difference in executive control functions between three different groups, namely, the expert shooters, novice shooters, and the non-experienced controls. The authors aimed to examine how the executive function, in this case, the suppressed processes, might differ from the participants with different expertise. The data exhibited the response rate was higher in the expert group than the control group, suggesting a better information processing and higher efficiency in the cortical connection. This manuscript is clear and concise, and the authors show a revealing command of future studies. The authors have done a great job on providing an informative and meaningful addition to the literature on the application of the insights of cortical processing regarding the expertise.

However, there are several changes that the authors are encouraged to revise to elevate the overall contribution of the paper to this research field.

Experimental design

Line 43-55. In this paragraph, the authors tried to explain the logical connections between the prefrontal cortex and the executive function. One missing link is missing here is the rationale of examining the logical relationships on closed sports. The readers might wonder why closed sports can also generate positive effects on executive function. Furthermore, is this beneficial effect a general influenced effect on motor performance or more referring to the task-specific phenomena?
2. Line 58. The authors might want to dig deeper in these references to secure the statement “training can increase α and β bands in the brain region” Are these references are cross-sectional studies or longitudinal studies?
3. Line 59-66. In this section, the authors tried to establish a relationship between sport training and cognitive functions. However, the processes to perform a motor behavior is complicated, and this session requires more evidence to support the rationale of investigating the cognitive functions and sport training. In addition, the authors may want to add more information to help explain the reason for using the shooting task.
4. Line 67-76. Again, the readers would be confused here as little supportive evidence to show the importance of using Flanker to probe the participants in three expertise categories.

Validity of the findings

5. Line 113. One concern regarding the manipulation check of the task is that did the experimenters ask the participants to keep the finger on the button when performing the task?
6. Results. The authors are recommended to report the effect size for the statistical analyses in this study.
7. Line 136-137. “The difference between the expert and novice groups was not significant for congruent trails, nor for incongruent trails” The readers may want to get to know the possible explanations in the discussion.
8. Line 165. The response time was also higher in the expert group when compared to the control group?
9. Line 170. The “intervention training” This term creates a noticeable confusion here. Theses references did not provide the intervention in the studies. The authors are recommended for providing more information regarding this issue.
10. Line 175. How the authors can state the results which the executive function contributed the shooting expertise? Specifically, why the performance of suppression serves as a critical factor here to be relating to the shooting performance?
11. Line 188. “Closed exercise” Do the authors try to refer the closed skill in sports?

Reviewer 2 ·

Basic reporting

Please see the general comments.

Experimental design

Please see the general comments.

Validity of the findings

The validity of the findings should be carefully checked by the authors.

Additional comments

General comments for the author

The current study examined the differences in executive function using a flanker task between shooting players with different expertise level, as well as non-athletic controls. Findings suggest task performance of the expert group were better that of the novice group. With these findings the authors concluded that long-term participation in shooting can improve executive function as well as neural efficiency. Without question, this type of work is needed in the field to better understand what cognitive and neural processes differentiate individual difference with different sports expertise level. However, the current manuscript would benefit from several conceptual, methodological, and reporting changes. The reviewer presents these thoughts and concerns below.

Abstract
More details of the main findings should be presented explicitly.

Introduction
The major concern is the examination of neural efficiency without recording any brain data, which may preclude any inference about neural processing. If any behavioral index has the capacity to demonstrate neural efficiency during cognitive processes, please state clearly with appropriate citations.

The rationale for this type of investigation is still unclear. More clear and stronger reasons are needed not just this study was conducted simply due to only few studies focusing on closed-skill sports.

Because this study specifically focused on shooting athletes, this reviewer suggests to provide more relevant literature about the cognitive and neural characteristics of this particular type of athletes, especially in those demonstrating neural efficiency in shooting athletes.

Line 56, please provide citations to support the statements here. I believe that not all research has failed to demonstrate the relationship between systematic training and neural efficiency during executive functioning.

Line 58 – 61, it seems likely that the EEG findings observed in these literature contract with the neural efficiency assumption, with the evidence showing greater EEG activities in professional shooters. Please clarify.

Please provide reasons regarding the choice of flanker task. For example, why the cognitive inhibition of executive function is crucial for examining the neural efficiency of expert shooter?

Line 67, because this study adopted a cross-sectional design, it may not be able to address the effect of training. Please modify the study purpose accordingly.

Line 73 – 76, the hypothesis is not clear here. Why the expertise effect may differ across incongruent and congruent conditions? Please provide citations to support this argument.

Methods & data processing
Please provide statistical analysis for the demographic data. One of the shortcomings is the age difference across groups that may potentially bias the effects of interest.

Since executive function may strongly vary with intelligence, education levels, depressive status, cardiorespiratory fitness, years of training experience, and so on. If any, these data should be presented, preferably in a sample characteristics table.

The procedure for data reduction should be clearly described.

This reviewer strongly suggest that the authors should check the data carefully. The data presented in the table seems very unreasonable (i.e., RT < 200ms or accuracy < 50%).

For the statistical analysis of RT, because the interaction between the two factors did not reach the significance level, it is not appropriate to examine the group effect for each condition, separately. Importantly, on the basis of the RT data presented in this study, it seems that the RT was faster for the controls relative to that of expert & novice (also see Table 1), which is completely inconsistent with the conclusion of this study (see abstract & discussion).

Discussion
Thus far, I feel unable to evaluate the discussion until the concerns above could be appropriate addressed.

Conclusion
The conclusion can not be supported by the data presented in this study.

---

## Round 0.2 · Major Revisions

I now have received two reviewers' comments. Although both reviewers are positive about your revision based on previous comment, they also pointed out some issues that need additional attention. Please revise or refute according to the reviewer’s comments and provide a point by point reply in addition to the revised manuscript.

Tsung-Min Hung, PhD., FNAK, FISSP
PeerJ editor
Research chair professor,
Department of Physical Education,
National Taiwan Normal University

·

Basic reporting

The manuscript has been improved significantly; the authors have my congrats. However, there are several changes that the authors are encouraged to revise to fine-tune the manuscript to meet the standard of publishing on PeerJ.

Experimental design

No further issues concerning the experimental design

Validity of the findings

Further information regarding the validity of the findings is listed in the general comments

Additional comments

1. Line 24-33. Although the authors provided further information on the main results of this study, the results provided here do not point out any conclusive evidence which supports the neural efficiency, which states on line 33.
2. Line 44. Please explain the findings of these cited research regarding the lower activation of the brain is associated with a higher skill level.
3. Line 51. This paragraph contains way too many contexts, leading to great difficulty in comprehending this information. The functional role of prefrontal cortex and the types of exercise are recommended to be separated into two paragraphs. Incidentally, what is the definition of open and close exercise?
4. Line 81. Do the authors try to state that shooting is part of close exercises here?
5. Line 90. How do the studies (from Hillman et al. and Hatfield) state that the shooting training can improve the brain waves? Are these studies cross-sectional or longitudinal?
6. Line 92. The authors are recommended to provide further information regarding these studies. And, how these studies can be related to the aim of this manuscript?
7. Line 96. There is no evidence, from the texts, showing that the neural efficiency is behind the findings cited in this paragraph. The readers would be confused about how these pieces of evidence can contribute to the hypotheses.
8. Line 104. The readers might wonder what the ACC refers to.
9. Line 150. The authors are encouraged to inform the readers about the types of effect size and the criteria used in this manuscript.

Reviewer 2 ·

Basic reporting

See the general comments

Experimental design

See the general comments

Validity of the findings

See the general comments

Additional comments

I think that the manuscript has considerably improved and I acknowledge that the authors have done a good job in taking care of the suggestions made by the reviewers. However, a few more issues need to be addressed before this manuscript moving toward publication.


1. The ages of participants seem to vary as a function of groups. This concern may be mitigated if the authors could include this factor in analyses as covariates. This concern is also worthy to be mentioned in the limitation section.

2. One major concern is that the conclusion regarding neural efficiency made here was not supported by the data, given the lack of brain measures in this study. Perhaps addressing the difference in domain-general cognitive functions may be more appropriate here.

3. It is suggested to provide the rationale for the design of ISI of the flanker task. Is this fact important for the investigation of expertise level here? Please clarify.

4. The factor of ISI should have three levels, please correct.

5. Finally, the major novel conclusion of this study, unfortunately, does not appear to be conclusively supported by the statistical evidence. To elaborate, given the lack of three-factors interactions, the simple main effects of group comparisons for each ISI in each trial type were not allowed. This issue should be carefully addressed because the weak statistical support for it is highly problematic.

---

## Round 0.3 · Major Revisions

I now have received the reviewer's comments on your revised manuscript. The reviewers have pointed out several aspect of your manuscript that still require substantial revision. Please revise or refute according to the reviewer's comments and provide a point by point reply in addition to the revised manuscript.

Tsung-Min Hung, PhD., FNAK, FISSP
PeerJ Editor
Research chair professor,
Department of Physical Education,
National Taiwan Normal University

·

Basic reporting

Given there's no evidence supporting the neural efficiency hypothesis, it is hard to connect all the points raised by the authors in this manuscript. Furthermore, after proofreading and the first revision, the quality of writing was expected to be improved extensively. However, many typos and abbreviations are not corrected.

Experimental design

no comment

Validity of the findings

In the introduction, there are four paragraphs talking about the relationship between motor behavior and executive function. However, when looking into the discussion, the tone was set to relate the results to the neural efficiency hypothesis. This contradiction is confusing. Hence, the inference in the direction of the effect experience on neural activity is relatively weak, as the dependent variables in this study did not provide the support to correlate the motor behavior (Flanker test) and executive function or neural efficiency (neural activity).

Additional comments

The results section is ambiguous. Besides listing the significant differences, the authors are recommended to point out the directions, e.g., the control group showed lower ACC than the novice group.

Reviewer 2 ·

Basic reporting

no comment

Experimental design

no comment

Validity of the findings

no comment

Additional comments

This reviewer appreciates the effort for the revision by the authors. I only have few minor issues that needs to be addressed before this manuscript moving toward publication.

1. In abstract, please explicitly indicate the main findings rather than just mentioning the group differences.

2. Please create a table showing all the statistical results for the ANCOVA tests. In this version, the authors did not report the statistical findings for the covariates.

3. The headings for the tables are incorrect. Please carefully check any potential mistakes.

---

## Round 0.4 · Minor Revisions

I have now received two reviewers’ comment and both reviewers were generally satisfied with your reply and revisions from previous comments. However, a few issues remained to be addressed before I can accept your manuscript. Please take care of these issues and provide a point by point reply in addition to the revised manuscript.

Tsung-Min Hung, PhD., FNAK, FISSP
PeerJ editor
Research chair professor,
Department of Physical Education,
National Taiwan Normal University

·

Basic reporting

The authors have done a great job of updating and revising the manuscript with a more informative and meaningful addition to the literature.

However, there are amendments that the authors are encouraged to revise to elevate the overall contribution of the paper to this research field.

General:
1. Abstract. In the conclusions, what do the authors mean precisely concerning “plasticity” in the current manuscript? In the sense of a cross-sectional study, how did the authors measure the shooters’ plasticity?
2. Terminology. Please be cautious regarding the difference between exercise and motor skills. One specific term used in this manuscript is particularly confusing, namely, exercise. For example, on line 12, the authors described the effects of open and closed exercise and their possible links to the executive function. However, this manuscript focused on the different experience on the motor skill, specifically, shooting, and its relation to the executive function. The authors are recommended to be clearer to use motor skills instead of exercise as the definition is not the same.

Experimental design

Please see the general comments below

Validity of the findings

Please see the general comments below

Additional comments

Specific comments:
1. Line 22. It might be nice to have a concluded statement here to finalize the studies mentioned above.
2. Line 23-33. Two disconcert concepts are listed in this paragraph. Line 23-31 is addressing the relationship between closed skill sports and the importance of executive function. However, line 31-33 is recommended serving as the title sentence of this paragraph as it might increase the reading flow of the readers.
3. Line 43. Self-repression. Please justify this term and may be a good idea to cite the article that supports this statement.
4. Line 45. Closed skill sport? The authors seem not clear to differentiate the concepts between exercise and sport. A precise proofread is recommended for the entire manuscript.
5. Line 45, the final paragraph in the introduction. The reader might wonder what prompted the authors to run this experiment as there is no research question was addressed.
6. Line 48. Regarding the Flanker task, the author might want to clarify what mental factors or constructs the Flanker task can measure and the rationale of using the Flanker task in the current study.
7. Line 62. Athletes. Do the authors mean novice shooters here?
8. Line 136. Please specify the significance of the main effect here.
9. Line 154. It might be better to start discussing the results of RTs in another paragraph to improve the overall readability of this paragraph.
10. Line 161. Do the authors mean “faster” response speed here?
11. Line 168. The authors claim in advance that two years of clay pigeon training is good enough to make the shooters from a novice to a professional shooter. Nevertheless, when conducted this experiment, the average training year of the novice group was 3.39 years. How do the authors persuade the readers that this was a well-controlled experiment?
12. Line 172. The authors try to discuss “again” the results of accuracy in this study. Would it be a good idea to move this part into the previous part (Line 146) to make the discussion session more streamlined?
13. Line 176. Please check the citation of Wang et al. (2013). Again, it should be closed or open skilled sports. Not exercise.
14. Line 185. How do the authors define the “plasticity” in this study? Have the authors measured the plasticity in this manuscript?

Reviewer 2 ·

Basic reporting

See the general comment.

Experimental design

See the general comment.

Validity of the findings

See the general comment.

Additional comments

The manuscript has been improved significantly. I only have some minor comments for further improving the manuscript.

1.The conclusion should be revised to be in line with the findings. I believe this study could not examine the characteristics of sports because only one type of sport was examined.

2.The English writing could be improved in order to enhance its readability of the manuscript.

3.Please correct the reference format.

---

## Round 0.5 · Minor Revisions

I now have received the reviewer's comments and there is one suggestion for you to consider. Please check and proofread the entire manuscript before I can make the final decision.

Tsung-Min Hung, PhD., FNAK, FISSP
PeerJ editor
Research chair professor,
Department of Physical Education,
National Taiwan Normal University

·

Basic reporting

After the revision, I have no further questions regarding the manuscript. There's one minor issue that the authors might want to revise is the in-text citation. For instance, line 13 (Tsai and Wang, 2015). The authors may want to check the format of in-text citations throughout the manuscript.

Experimental design

The experimental design is now clear and well-addressed.

Validity of the findings

No further questions regarding the findings.

---

## Round 0.6 · accepted · Accept

I have read through your reply to the reviewer's comment and your revised manuscript. I am satisfied with your response and decided that there is no need to send to the reviewer. You and your coauthors have my congratulations. Thank you for choosing PeerJ as a venue for publishing your research work and I look forward to receiving more of your work in the future.

Tsung-Min Hung, PhD., FNAK, FISSP
PeerJ editor
Research chair professor,
Department of Physical Education,
National Taiwan Normal University